# Waist Circumference as a Preventive Tool of Atherogenic Dyslipidemia and Obesity-Associated Cardiovascular Risk in Young Adults Males: A Cross-Sectional Pilot Study

**DOI:** 10.3390/diagnostics10121033

**Published:** 2020-12-02

**Authors:** Alberto Hernández-Reyes, Ángela Vidal, Alicia Moreno-Ortega, Fernando Cámara-Martos, Rafael Moreno-Rojas

**Affiliations:** 1Department of Bromatology and Food Technology, University of Córdoba, 14071 Córdoba, Spain; t22moora@uco.es (A.M.-O.); fernando.camara@uco.es (F.C.-M.); rafael.moreno@uco.es (R.M.-R.); 2Department of Animal Medicine and Surgery, University of Córdoba, 14071 Córdoba, Spain; v92vicaa@uco.es

**Keywords:** atherogenic dyslipidemia, waist circumference, central adiposity

## Abstract

Although the correlation coefficient between body mass index (BMI) and poor lipid profile has been reported, representing a cardiovascular risk, the need to find new early detection markers is real. Waist circumference and markers of atherogenic dyslipidemia are not usually measured in medical review appointments. The present study aimed to investigate the relationship between central adiposity and cardiovascular risk. This was a cross-sectional pilot study of 57 young males (age: 35.9 ± 10.85, BMI: 32.4 ± 6.08) recruited from community settings and allocated to non-obese or obese attending to their waist circumference. Total cholesterol (TC), high-density lipoproteins (HDL-C), and low-density lipoproteins (LDL-C) cholesterol and triglycerides (TG) were measured from plasma samples. Patients with at least 100 cm of waist circumference had significantly increased TC, LDL-C, non-HDL-C, and triglycerides and lower levels of HDL-C. The three atherogenic ratios TC/HDL-C, LDL-C/HDL-C, and TG/HDL-C were all optimal in non-obese patients. LDL-C/HDL-C and TG/HDL-C were significantly higher and over the limit when assessing for atherogenic dyslipidemia. The number of patients at risk for cardiovascular events increases 2.5 folds in obese compared to non-obese. Measurement of waist circumference could be adopted as a simpler valid alternative to BMI for health promotion, to alert those at risk of atherogenic dyslipidemia.

## 1. Introduction

Obesity is spreading across the planet at an unstoppable rate. It affects not only developed countries but also lower-middle-income countries, jeopardizing the health systems’ viability [1]. In Spain, overweight and obesity have a prevalence equal to 61.6%, occupying the 18th position concerning the ranking of 52 Organisation for Economic Cooperation and Development (OECD) countries [2]. The leading causes of mortality in the world are ischemic heart disease and stroke, which caused 15.2 million deaths in 2016 and have been the principal causes of mortality in the last 15 years [3]. The concurrence of cardiovascular disease (CVD) in obese people is well established [4,5]. Metabolic syndrome is defined as cardiovascular (CV) risk related to obesity.

The classification of overweight or obesity according to the body mass index (BMI) has been widely used [6], even though body fat percentage (BFP) [7] or waist circumference (WC) [8] are beginning to be considered as novel markers in clinical research for the diagnosis of obesity.

The main problem with using BMI in isolation to measure CVD is due to an insufficient biomarker of central adiposity (CA) [9]. WC is a simple method to measure CA, easy to apply, and standardized in clinical practice. WC is strongly associated with CV mortality [10], with or without adjustment for BMI [11]. Focusing the study on WC as a predictor for the risk of suffering a CV event could provide more accurate opportunities in the diagnosis and evaluation of the treatment of the patient with obesity and related CVD.

Shen et al. reported that WC shows a greater association with metabolic syndrome components than BFP or BMI [12]. A high WC is a significant independent contributor to high levels of triglycerides (TG) [13], as well as to low plasma high-density lipoprotein cholesterol (HDL-C) concentrations [14]. Various WC thresholds have been accepted to assess the health risks related to abdominal and visceral obesity [15,16]. Given that certain CV risk markers’ prevalence increases proportionally with adiposity [17], Ardern et al. have proposed specific WC thresholds according to sex and BMI categories to more efficiently identify people at higher risk for CVD [18]. The Consensus Statement of the International Atherosclerosis Society (IAS) and International Chair on Cardiometabolic Risk (ICCR) Working Group on Visceral Obesity proposes specific thresholds for overweight people to sex, WC ‡ 100 cm for men. [9].

For the study of the diagnosis and prevalence of CV events, we have, among other markers, atherogenic dyslipidemia (AD). AD is characterized by an elevation of TG and reduced HDL-C levels and has been reported to strongly predict CV morbidity, especially coronary artery disease in the general population [19,20]. Along with these two lipid alterations, which define AD, it is likely to find a moderate increase in cholesterol concentrations of low-density lipoproteins (LDL-C) [21]. AD is associated with highly prevalent pathologies in the general population and represent a high CV risk, such as obesity, overweight, and metabolic syndrome [22].

The observational evidence confirms the importance of having, from an early stage of adulthood, high LDL-C levels, and the correlation with a future cause of CVD is strong. [23]. Few studies directly link atherosclerosis observed in young people postmortem to cholesterol levels measured premortem or postmortem [24]. Trajectories in that cohort started at a mean age of around 45 years. Although preventive drug therapies reduce the relative risk of CV events in primary and secondary prevention patients, the absolute risk of subsequent CVD events remains high [25]. It is estimated that, in the year 2035, almost half of the US population will have some CVD [26].

To the best of our knowledge, the present study is the first to describe, in the same sample, the relationships between WC, anthropometric characteristics, and lipidic biochemical parameters in young adult men. It is noteworthy that most of the studies carried out previously classify patients with CV risk according to the plasma values of TC, TG, HDL-C, and LDL-C, but the fact is that CV risk differs considerably depending on the parameter evaluated.

Thus, new systematic approaches are necessary to improve lifestyle habits and also better control in clinical diagnosis to identify possible risk factors early (Figure 1).

Along with the aforementioned alterations, the increase in atherogenic coefficients, especially total cholesterol (TC)/HDL-C and TG/HDL-C, form the set of markers for the diagnosis of this pathology, summarized in Table 1 [27].

This study hypothesized that more prominent CA would be associated with atherogenic dyslipidemia in the CV risk population, such as overweight/obese young adult males. To this purpose, we examined the association of WC and BFP with atherogenic dyslipidemia-associated lipid profile and studied its relation to evaluating the CV risk in both obese and non-obese men.

## 2. Materials and Methods 

### 2.1. Study Design 

This study was a cross-sectional survey. A sample of 57 adult males, with a mean age of 35.9 ± 10.85 years and Caucasian origin, were recruited from several sources, including an obese clinic, dieticians, and from responding to a social media release. Recruitment took place in Cadiz city and surroundings, Andalusia. All men attended a face-to-face interview to evaluate the inclusion criteria for the study. Data about demographic factors (e.g., age, sex, and education), lifestyles (e.g., smoking, alcohol consumption, and physical activity), cardiovascular risk factors (e.g., obesity, hypertension, and diabetes), use of medications were collected through interviews. Clinical examinations and laboratory tests were performed on the recruited patients to obtain their anthropometric characteristics and lipid profile. This study was approved by the bioethical committee of Córdoba University, in the Department of Health at the Regional Government of Andalusia (Act n°284, ref.4156). Participants provided written consent for their participation.

The recruited subjects were allocated to a group based on the obesity condition. ‘Non-obese’ and ‘obese’ groups were defined, attending either BFP or WC, and anthropometric variables and biochemical analysis were recorded for statistical analysis. The percentage of subjects of the limit for atherogenic dyslipidemia was calculated to evaluate the probability of the population for CVD related to obesity.

Abdominal obesity was defined as BFP ≥28% [28]. Therefore, for studies related to BPF, participants were allocated to the ‘non-obese’ group when body fat was less than 28%; likewise, men with 28% or more were classified as ‘obese.’

Abdominal obesity was defined as a WC ≥100 cm in men by the Consensus Statement from the IAD and ICCR Working Group on Visceral Obesity [9]. Thus, ‘non-obese’ patients were considered when WC <100 cm and ‘obese’ for equal or higher measure. The influence of the obesity evolution was assessed by layered the obese patients into three different groups according to the scale of WC propose by Ardem et al. [18].

### 2.2. Inclusion/Exclusion Criteria

The health condition of each subject was determined by evaluating the data collected after a personal interview. Every medium-aged healthy man with or without obesity was included in the study. Concomitant diseases like renal dysfunction, hypertension, or diabetes were established as exclusion criteria. Any treatment, either temporal or chronic, was avoided during the recruitment process since some drugs could introduce confusing variables in the study of metabolic profile and cardiovascular function. Sedentary physical activity (determined by personal electronic devices) was established as preferential and to avoid moderate-high and intense.

### 2.3. Data Collection and Sampling Procedure

For the height, a stadiometer was used (SECA 213) without shoes to the nearest 5 mm, with the head positioned so that the eye and the external auditory meatus were leveled. BFP and muscle mass (MM) were recorded by multifrequency electrical impedance (BWB-800A, Tanita Corp., Tokyo, Japan), previously validated [29]. This method is based on a 3-compartment model capable of evaluating BFP, MM, and bone mineral content. The independent variables recorded were: age (years), height (m), weight (Kg) and BMI, which was calculated as weight (kg)/[height (m)^2^]. Waist circumference was measured at the midpoint between the lowest rib and the iliac crest (in centimeters) [30].

Participants attended a screening consultation after an overnight fast. Fasting blood samples of 10 mL were collected from each patient, and serum was separated by immediate centrifugation at four °C and 3500 rpm for 10 min. TC, TG, and HDL-C were analyzed by enzymatic CHDD-PAP, GPD-PAP assays, and homogenous methods. A25 auto analyzer, BioSystems, USA, was employed. LDL-C levels were calculated using the Friedewald method [31]. After obtaining the entire lipid profile, the coefficients for TC/HDL-C, LDL-C/HDL-C, and TG/HDL-C were calculated from the corresponding lipids’ levels.

All study data collected were obtained by a trained and experienced dietitian, and a laboratory test was performed by the qualified personnel in the Laboratorio Vidal Zambrano, Chipiona, Cádiz, Andalusia.

### 2.4. Cardiovascular Risk Assessment

The lipid profile ranges and ratios commonly used to diagnose atherogenic dyslipidemia were used as indirect markers to assess cardiovascular risk considering the CA measured by WC and BFP. The percentage of patients with the lipid level out of range was calculated for each parameter measured in non-obese and obese situations. The evaluation of the prevalence determined the CV risk. A study correlation was performed to determine the role of CA markers in the prediction of the severity of lipid disorders in obese and non-obese.

### 2.5. Statistical Analyses

The power calculation was not performed because this study is exploratory to inform future studies’ design. All statistical procedures were performed using GraphPad Prism software version 6.01 (GraphPad Software, La Jolla, CA, USA). Values are expressed as the mean ± standard deviation (SD). The difference between means for two different groups was determined by *t*-test; one-way ANOVA assessed the difference between means for three or more groups. All variables were measured on a continuous scale, and no transformations were made because they presented a normal distribution. Fisher LSD test was used as a post-hoc procedure. Chi-square test has been used to compare proportions between groups. Receiver Operating Characteristic curves have been carried out to determine the cut-off values. The correlation study was carried out using the Pearson test. Statistical significance was considered when *p* < 0.05.

## 3. Results

STROBE reports the results (Strengthening the Reporting of Observational Studies in Epidemiology). See Appendix A for the STROBE checklist [32]. A total of 57 young adult men were eligible to participate in the study. The characteristics of the study sample are summarised in Table 2 and Table 3.

### 3.1. Body Fat Percentage and Lipid Profile Evaluation

Lipid biochemistry was evaluated for each patient according to an obesity classification based on total BFP. Anthropometrics characteristics for the obese and non-obese groups are shown in Table 4.

As shown in Table 5, when BFP was used for classification, worst lipid profile was observed in obese men. Nonetheless, no significant differences were found in most of the parameters evaluated when compared to those considered non-obese. Only LDL-C cholesterol (*p* = 0.0254) and LDL-C/HDL-C ratio (*p* = 0.0331) was significantly increased in the obese group. Non-obese men had most of the lipid parameters in normal levels although LDL-C and TG/HDL-C ratio were upper the established limit. In the obese group, we also found normal levels of triglycerides, HDL-C and TC/HDL-C ratio but LDL-C and TG/HDL-C ratio were also upper limit. In addition, obese men had increased non-HDL-C and LDL-C/HDL-C values.

### 3.2. Waist Circumference and Lipid Profile Evaluation

The WC measure was cut-off in 100 cm to evaluate lipid changes according to obesity classification. Therefore, each patient with a value equal or higher than 100 cm was considered obese. Anthropometrics characteristics for the obese and non-obese group according to waist circumference are shown in Table 6.

Although BFP and WC were well correlated in our participants (r = 0.8538, *p* < 0.0001) when the cut-off for obesity was carried out by using the WC instead the total body fat, a lot of differences between obese and non-obese group were observed, as shown for body fat, lipid profile gets worse in obese men. However, in contrast to BFP, WC’s use in obesity classification showed significant differences for each parameter evaluated (Table 7). We also observed an increase in the ratios commonly used to evaluate atherogenic dyslipidemia. Again, these three parameters were significantly higher in men with less than 100 cm of WC. In this context, although the obese group had normal values for triglycerides, HDL-C, and TC/HDL-C, non-obese men did not present any disruption on any of each measure.

### 3.3. Study of Obesity Evolution on Lipid Profile

When the obese group was layered according to the WC into the exact grade of overweight, the 57 men were assigned to one of the following four groups: non-obese (WC < 100 cm), overweight (WC from 100 to 109 cm), obese I (WC from 110 to 124 cm), obese II (WC ≥ 125 cm). The evolution in lipid profile concerning obese status is shown in Table 8. TC, LDL-C, and non-HDL-C were increased in the three obese groups, but there were no differences along with obese status. In contrast, HDL-C tended to decrease, and TG increased in the progression of obesity with a marked change in the obese II group. The atherogenic lipid ratios were consistently higher in the three obese groups and increased with the grade of obesity.

### 3.4. Cardiovascular Risk Assessment Associated with Atherogenic Dyslipidaemia

Each parameter of the lipid profile was evaluated individually to analyze the CV risk associated with obesity. The percentage of patients with the probability to develop CV events according to cholesterol and triglycerides levels was higher in the obese group either when we used BFP (Figure 2A) or WC (Figure 2B) to classify patients. Although the risk was almost the same in obese, a notable difference was found when non-obese men’s risk. This percentage increased 2-fold when used body fat against to circumference measure. It is noteworthy that LDL-C levels exceeded the limit in more than 41.7% of patients even when they were not overweight and increased up to 75% when obesity was present. We also found that sorting patients according to WC, reduced to 16.7% from 36% the number of non-obese men at risk when analyzing TC levels, 16% to 8.3% for HDL-C, and 40% to 25% for non-HDL-C.

Regarding atherogenic ratios, when patients were classified as obese or not obese based on the BFP, TC/HDL-C, and LDL-C/HDL-C ratios, it showed that up to 16% and 12% of non-obese patients, respectively, were at risk for CVD, increasing in obese men (Figure 2A). In contrast, when WC was the cut-off for obesity, these parameters showed that non-obese patients were potentially at no risk for CV events, but this number increased to 18–27% in obese men (Figure 2B). Unexpectedly, in both obesity classifications, the TG/HDL-C ratio was the upper limit in a considerable percentage of men (44% and 33% respectively), even when they were considered non-obese. Again, this parameter increased up to 53% in obese.

Concerning the CV risk evaluation, when a more accurate stage for obesity was assessed considering waist circumference, we observed that an increase in the grade of obesity also implied an increase in the percentage of patients at risk in most of the lipid parameters profile. Although light changes were observed between the overweight and obese I group, the percentage of men increased an average of 20% in both when compared to risky but non-obese men. This percentage increased an additional 20% to the obese II group, which means around a 40% higher of patients with CV risk in the group with grade II of obesity concerning non-obese (Figure 2C).

Correlation studies showed that BFP and WC were positively correlated with LDL-C and non-HDL-C levels and had a negatively correlated with HDL-C levels. The atherogenic ratios TC/HDL-C and LDL-C/HDL-C increased with WC and BFP, and TG/HDL-C also showed a positive correlation with WC (Table 9).

When we divided patients into non-obese and obese groups, we found strong correlations between adiposity (WC or BFP) and cholesterol parameters and their ratios in non-obese men. These values were higher when using WC (Figure 3A–E) than BFP (Figure 3F–J). However, in the obese group, we did not find any correlations between lipid profile with WC nor BFP.

## 4. Discussion

It is widely accepted that obesity induces alterations in lipid biochemistry, evolving towards dyslipidemic atherogenesis, a critical factor in the development of cardiovascular events. In this context, the early diagnosis of obesity is an essential clinical goal in preventing CVD. This study was designed to evaluate the diagnostic efficacy of adiposity measurement, using body fat and waist circumference, as a tool for the early detection of obesity and associated lipid disorders. Our results demonstrate that the use of BFP cannot detect both obesity and changes in men’s lipid profile at the same time. Only the LDL-C values had a significant difference between obese and non-obese. However, both groups presented these levels exceeded.

In contrast, WC’s use showed significant differences in all the parameters of the lipid profile parameters when obese and non-obese patients were compared. Furthermore, all lipid values were within normality in the non-obese group, and most of them altered, or almost, in obese patients. Thus, even though either BFP or WC can be an excellent tool to diagnose obesity in the context of dietary treatment, WC is more reliable to also diagnose lipid disruption associated with obesity. Moreover, when we sort patients according to the lipid limits established for dyslipidemic atherogenesis, WC was more accurate to relate to obesity and CV risk.

CVD is the primary cause of morbidity and death in Western Societies [32]. In the general population, dyslipidemia and obesity have been identified to be predisposing factors for diabetes mellitus and increase CV risk in those patients [33]. Moreover, in women, the need for accurate diagnosis of obesity has been elucidated since it considerably increases the risk of CVD, and treatment of dyslipidemia may be challenging [34]. It is known that CVD pathogenesis differs concerning sex and that hypertrophic changes in the left ventricle in men are strongly associated with CV mortality [35]. Thus, despite the general population should be assessed, we focus this study on males.

Although it continues to associate BMI, TC, TG, and LDL-C with the risk of suffering an event CVD [36,37,38,39,40], we did not find an association between BMI, and TC or TG correlation between BMI and LDL-C was weak (data not shown). Although clinical trials provide the highest level of evidence for the value of cholesterol reduction, the results of these studies suggest that these interventions may be too little and too late. Lipid-lowering assays tend to be conducted in high-risk populations, so differences in event rates can be detected at relatively short time intervals [23].

It was relatively surprising that neither TC nor TG was significantly correlated with BMI in this study. The recent prospective study of Kwon et al. [41] carried out in 503,340 Koreans concluded that “It remains unclear whether the lowest cholesterol levels are associated with the least mortality from CVD and IHD in Korean adults.” Most studies did not evaluate the association between change in cholesterol levels and CVD outcomes. Although many previous studies have evaluated the effect of baseline cholesterol on CVD risk or mortality, the population size was small and limited to men [42,43]. Although high TC in men is known to be a predictor for CV events, there is not enough evidence for the association of cholesterol level change with CVD. In the recent work of Su-min Jeong et al. [44], it was evaluated whether the change in cholesterol is associated with CVD incidence among young adults. The results, after examining more than 2.5 million young adults (aged 20–39 years) who had undergone two consecutive national health check-ups provided by Korean National Health Insurance Service between 2002 and 2005, indicate that increased cholesterol levels were associated with high CVD risk in young adults. However, cholesterol levels were considered high when ≥240 mg/dL, a value higher than that used for the diagnosis of hypercholesterolemia.

The three abnormalities of raised serum TG, increased LDL particles, and decreased HDL have been termed the atherogenic lipoprotein phenotype or, more simply, the lipid triad [45]. This multiplex array of lipid abnormalities is a potent risk factor for CVD.

According to CA, those subjects classified as non-obese would be considered risk-free when considering the triglyceride value, TC/HDL-C ratio, and LDL-C/HDL-C ratio, and only 8% of these men would be at risk of CVD depending on the level plasma HDL-C. In contrast, when TC, LDL-C, non-HDL-C, or TG/HDL-C ratio is individually assessed, up to 40% of non-obese men are potentially sensitive to CV risk. These percentages are even higher when the parameter used to discriminate between obese and non-obese is BFP. In this case, our results show that the average in normal weight with CV risk rises to 56% based on the LDL-C value and around 40% for TC, non-HDL-C, and TG/HDL-C ratio. All these data suggest that the absence of obesity is not equal to the absence of lipids disruption and associated-CV risk. This may be explained because non-obese people also have the metabolic disease [46], which could deteriorate CV function. In this context, WC seems to be a precise marker of obesity that discriminates against those with non-risk.

The results of our study show that at least 60% of men with overweight or obesity have average values of TC and HDL-C, 80% have regular TG, and up to 30% do not have LDL-C levels over the limit, so they would be classified as low or no risk people for CVD. This variability in the obese group is a direct consequence of the different degrees of obesity (from merely overweight to obese type II) and could be controversial when we pretend to inform our patients about the risk of dyslipidemia. Thus, if we consider the relationship between WC and AD markers, this percentage of men who would be considered not at risk for having a CV event should be evaluated according to the obese status and reconsidered their CVD risk because of this pattern.

It is remarkable that the percentage of patients at risk when analysing the lipid profile showed that both non-obese and obese had a certain probability of developing CV events associated with lipid disruption. However, WC classification reduced to half this number in non-obese patients. Thus, double of patients would be wrongly sorted when pretending to evaluate obesity-associated CV risk attending the standard BFP threshold. Also, correlation studies in this work showed that, even when CA may have a weak correlation with lipid parameters in the general population (Table 9), CA’s predictor effect in the severity of lipid disorders is strong in non-obese men (Figure 3). Thus, in the prevention of obesity to reduce the development of atherogenic dyslipidemia associated with this condition, CA seems to be a reliable tool. These correlations were more robust when we allocated patients according to WC instead of BFP. These data suggest that measuring and controlling the WC from early adulthood could be a marker with sufficient validity to initiate a drastic change in lifestyle, through dietary change and physical activity, as a preventive means to avoid suffering in the future an obese-associated CV event.

Considering the anthropometric characteristics of the patients studied, a BFP percentage of 28% or less may correspond to a WC of 100 or higher. However, the possibility that a man with <100 cm WC exceeds the threshold of 28% of BFP is null. This means that, although BFP may be useful as a first obesity screen, the WC measurement should be used to assess CV risk exposure, thus reducing the margin of error in diagnosis. A new optimal cut-off point should be considered to determine the percentage of body fat that best classifies the degree of obesity in adult men. Considering our data, with a ROC analysis, we found this point at a value of 23.4% of total body fat (AUC: 0.987, 95% CI 0.9648 to 1.009).

Our results could have important public health and clinical significance. It has been known for some time that plasma cholesterol is one of the main risk factors, so its control has become a preventive measure of the first order [47]. There has been a specific group of medications for decades: lipid-lowering or anti atheromatous drugs, intending to reduce total cholesterol and low-density lipoprotein levels [47]. The treatment with all these drugs is generally maintained indefinitely since the cholesterol level is suspended and returns to the pre-treatment levels. Two problems have existed around the prescription of this drug as a first measure due to a high CT: the high health cost involved [48], and it should always be, after prescribing physical activity and dietary treatment [49]. Although statin use rates are high in the highest LDL cholesterol groups, this use does not reduce event rates to those observed in the lifetime pathway group with the lowest cholesterol levels, potentially attributed to exposure. Before treatment at high cholesterol levels, it is safe to say that statins, as currently used, are not providing maximum impact as preventive treatments [23].

This fact is of clinical significance and, in our concern, future studies that propose evaluating CV risk in the adult male population should consider atherogenic ratios and body composition, including BFP and, WC and not isolated lipids parameters. Diagnosis through WC is a cheap, fast, non-invasive method and can be a marker that warns the patient of a future problem. As we have seen in this pilot study, if the criterion of suffering CVD risk is made through markers such as AD, it may give a new focus in the diagnosis and prescription of pharmacology. The reduction of morbidity in many cases is not expected after pharmacological treatment [50]. The CV event is, multifactorial and the effects of the drugs are, in turn, multiple. When establishing common LDL-C objectives based on epidemiological data, all the drugs, and circumstances due to the effects on final clinical variables [51]. Because AD usually precedes the clinical manifestation of the CVD, strategies to treat it focus on pharmacologic intervention. The results derived from this pilot work pretend elucidated how motorizing adiposity by WC could identify the AD-associated easily to obesity.

More studies should be considered with a higher number of participants to allow extrapolation of data. Moreover, the age included in this study does not represent how WC could affect aged population men, so we could not evaluate the role that plays the concomitant diseases in WC’s predictor effect. Also, longitudinal studies need to be performed to elucidate the influence of time, habits, and aging that could affect WC’s efficiency in anticipating changes in lipids profile.

## 5. Conclusions

Our results show that considering the lipid parameters established for assessing CV risk based on the diagnosis of AD individually, several overweight subjects are considered CV risk patients. However, a considerable number of overweight patients have a normal profile lipid. Furthermore, obesity needs to be ranged to accurately assess dyslipidemia. The diagnosis of obesity by using the BFP does not allow classifying those patients free of risk as non-obese people, while WC is a reliable measure to determine with greater precision the CV risk associated with obesity. Adiposity measured by WC also identifies those men potentially free of CV risk and classify them appropriately as non-obese men based on their lipid profile. Further works should aim to identify other contributory factors.

## Figures and Tables

**Figure 1 diagnostics-10-01033-f001:**
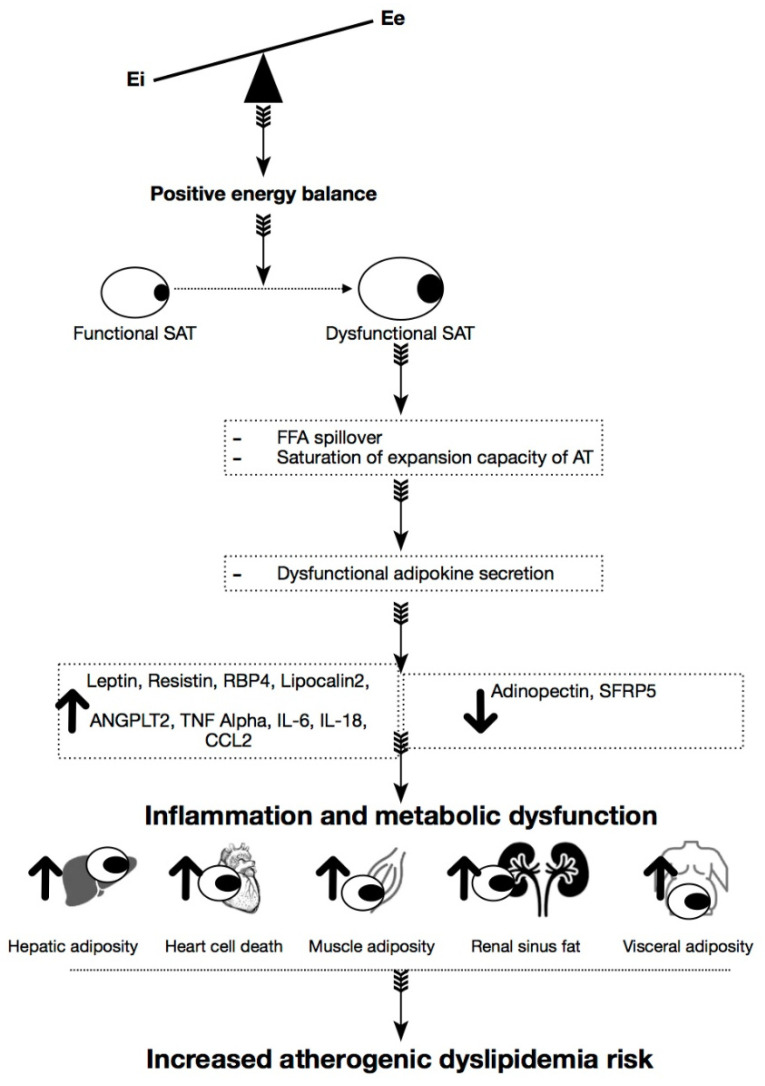
Overview of the potential role of excess visceral adiposity contributing to increased Atherogenic dyslipidemia risk. This figure illustrates the hypothesis that an uninterrupted positive energy imbalance over time can convert perfectly functional subcutaneous tissue cells into a generation of new fat cells (hyperplasia), allowing the storage of energy surplus. Over time, this adipose tissue becomes dysfunctional due to the subcutaneous Adipose tissue’s incapacity to expand and therefore saturate it. Under such circumstances, altered secretion of adipokines inflammatory cytokines occurs, among which we highlight: Leptin, Resistin, RBP4, Lipocalin2, ANGPLT2, TNF, IL-6, IL18, CCL2. This state of inflammation is maintained, increasing the ratios of dyslipidemic atherogenesis and causing harmful cardiometabolic consequences.

**Figure 2 diagnostics-10-01033-f002:**
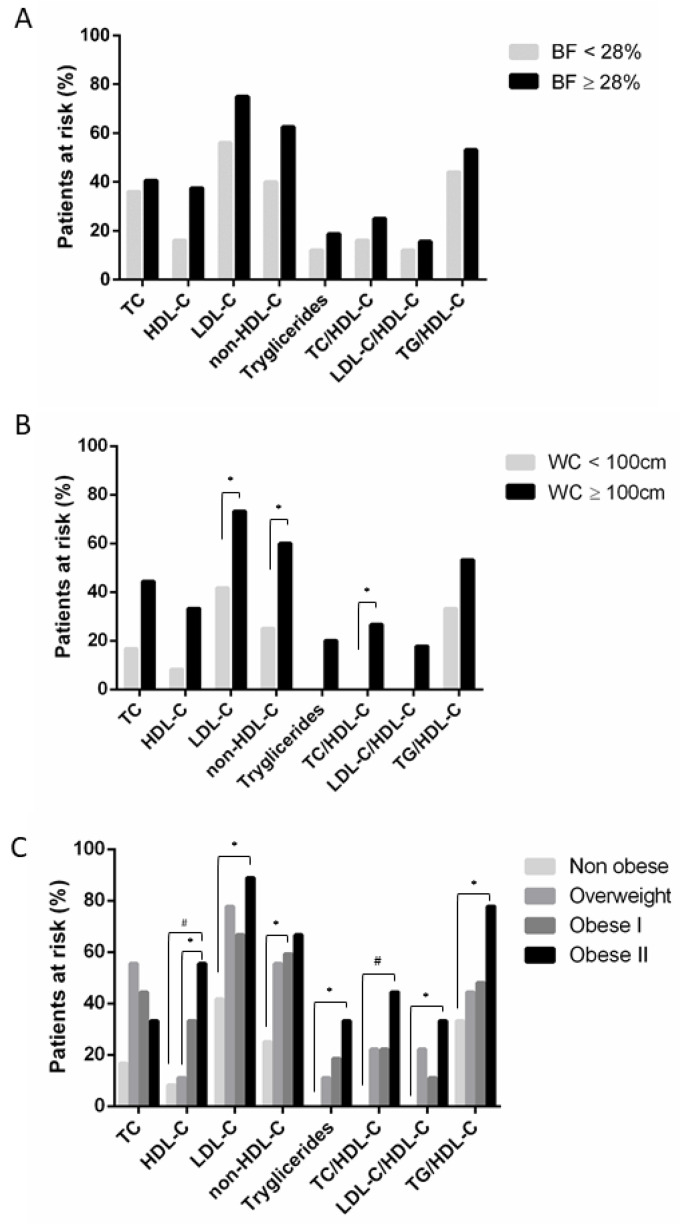
Cardiovascular risk evaluation attending to body fat percentage (BF, panel (**A**)), attending to waist circumference (WC, panels (**B**,**C**)). Each bar shows the percentage of patients at risk for cardiovascular events depending of lipid profile. HDL-C: high-density lipoprotein cholesterol; LDL-C: low-density lipoproteins cholesterol; TC: total cholesterol; TG: triglycerides; Significance is indicated when *p* < 0.05 (*) or *p* < 0.01 (#).

**Figure 3 diagnostics-10-01033-f003:**
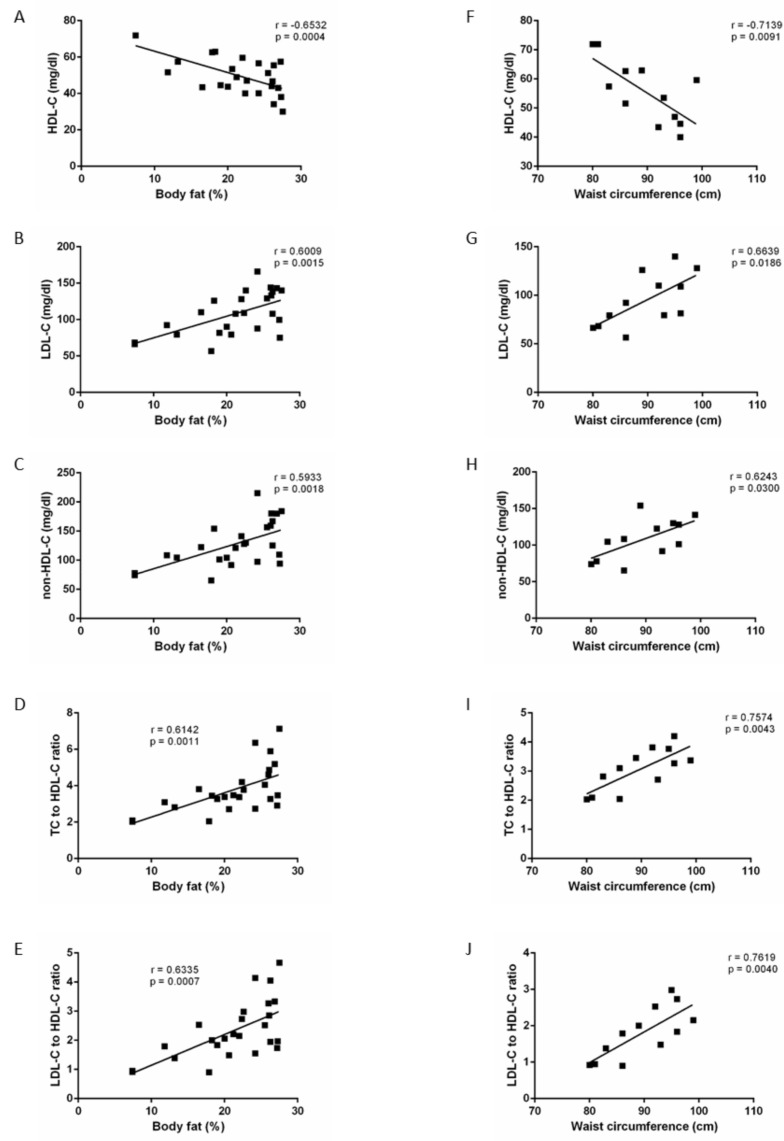
Correlation study (Pearson test) between body fat percentage (panels (**A**–**E**)) or waist circumference (**F**–**J**) and lipid parameters in non-obese men. HDL-C: high-density lipoprotein cholesterol; LDL-C: low-density lipoproteins cholesterol; TC: total cholesterol.

**Table 1 diagnostics-10-01033-t001:** Atherogenic dyslipidemia.

Biomarkers and Ratios	Values
TG	>150 mg/dl
HDL-C	<40 mg/dl
LDL-C	>100 mg/dl
non-HDL-C	>130 mg/dl
TC/HDL-C	>5
TG/HDL-C	>2
LDL-C/HDL-C	>2.5

HDL-C: high-density lipoprotein cholesterol; LDL-C: low-density lipoproteins cholesterol; TC: total cholesterol; TG: triglycerides.

**Table 2 diagnostics-10-01033-t002:** Anthropometric characteristics of the study sample.

Variable	Total (*n* = 57)
Age (years)	35.9 ± 10.85
Height (m)	1.77 ± 0.07
Weight (Kg)	101.2 ± 19.02
BMI (Kg/m^2^)	32.4 ± 6.08
Body fat (%)	27.4 ± 7.13
Muscle mass (Kg)	68.8 ± 8.38
WC (cm)	112.2 ± 15.90

BMI: body mass index; WC: waist circumference. Values are expressed as mean ± SD.

**Table 3 diagnostics-10-01033-t003:** Lipid profile and atherogenic ratios of the study sample.

Variable	Total (*n* = 57)
Triglycerides (mg/dl)	105.4 ± 56.58
Total Cholesterol (mg/dl).	185.6 ± 42.15
HDL-C (mg/dl)	48.1 ± 11.87
LDL-C (mg/dl)	118.9 ± 33.09
non-HDL-C (mg/dl)	137.5 ± 40.41
TC/HDL-C ratio	4.02 ± 1.11
TG/HDL-C ratio	2.39 ± 1.54
LDL-C/HDL-C ratio	2.60 ± 0.89

HDL-C: high-density lipoprotein cholesterol; LDL-C: low-density lipoproteins cholesterol; TC: total cholesterol; TG: triglycerides. Values are expressed as mean ± SD.

**Table 4 diagnostics-10-01033-t004:** Anthropometrics characteristics for the obese and non-obese groups according to body fat percentage.

Variable	Non-Obese (*n* = 25)	Obese (*n* = 32)	*p*-Value
Age (years)	33.1 ± 9.63	38.1 ± 11.4	0.0854
Height (m)	1.77 ± 0.07	1.76 ± 0.06	0.5188
Weight (Kg)	88.9 ± 15.27	110.8 ± 16.03	<0.0001
BMI (Kg/m^2^)	28.3 ± 4.76	35.7 ± 4.98	<0.0001
Body fat (%)	21.1 ± 6.01	32.2 ± 2.80	<0.0001
Muscle mass (Kg)	66.0 ± 8.61	71.1 ± 7.59	0.0212
WC (cm)	100.7 ± 12.68	121.3 ± 11.84	<0.0001

BMI: body mass index; WC: waist circumference Values are expressed as mean ± SD. The *t*-test analysis was performed to compare differences between groups. Statistical significance was considered when *p* < 0.05.

**Table 5 diagnostics-10-01033-t005:** Lipid profile based on the classification as obese or not according to body fat percentage.

Variable	Non-Obese (*n* = 25)	Obese (*n* = 32)	*p*-Value
Triglycerides (mg/dl)	100.1 ± 58.97	126.2 ± 55.23	0.5336
Total Cholesterol (mg/dl)	178.0 ± 33.41	191.5 ± 47.56	0.2320
HDL-C (mg/dl)	50.2 ± 10.78	46.4 ± 12.57	0.2356
LDL-C (mg/dl)	107.9 ± 29.64	127.5 ± 33.54	0.0254
non-HDL-C (mg/dl)	127.7 ± 38.55	145.1 ± 40.78	0.1089
TC/HDL-C ratio	3.76 ± 1.31	4.22 ± 0.90	0.1243
TG/HDL-C ratio	2.23 ± 1.71	2.51 ± 1.40	0.4979
LDL-C/HDL-C ratio	2.32 ± 1.00	2.82 ± 0.72	0.0331

HDL-C: high-density lipoprotein cholesterol; LDL-C: low-density lipoproteins cholesterol; TC: total cholesterol; TG: triglycerides. Values are expressed as mean ± SD. The t-test analysis was performed to compare differences between groups. Statistical significance was considered when *p* < 0.05.

**Table 6 diagnostics-10-01033-t006:** Anthropometrics characteristics for the obese and non-obese group according to waist circumference.

Variable	Non-Obese (*n* = 12)	Obese (*n* = 45)	*p*-Value
Age (years)	30.7 ± 8.73	37.3 ± 11.0	0.0605
Height (m)	1.78 ± 0.08	1.76 ± 0.06	0.3523
Weight (Kg)	78.1 ± 8.92	107.4 ± 16.01	<0.0001
BMI (Kg/m^2^)	24.5 ± 2.06	34.5 ± 4.95	<0.0001
Body fat (%)	16.6 ± 5.47	30.2 ± 4.14	<0.0001
Muscle mass (Kg)	61.5 ± 6.86	70.8 ± 7.68	0.0004
WC (cm)	89.7 ± 6.42	118.3 ± 11.64	<0.0001

BMI: body mass index; WC: waist circumference; Values are expressed as mean ± SD. The t-test analysis was performed to compare differences between groups. Statistical significance was considered when *p* < 0.05.

**Table 7 diagnostics-10-01033-t007:** Lipid profile based on the classification as obese or not according to waist circumference.

Variables	Non-Obese (*n* = 12)	Obese (*n* = 45)	*p*-Value
Triglycerides (mg/dl)	75.3 ± 33.84	113.5 ± 58.94	0.0366
TC (mg/dl)	163.8 ± 25.02	192.4 ± 44.06	0.0434
HDL-C (mg/dl)	55.5 ± 10.71	46.1 ± 11.46	0.0131
LDL-C (mg/dl)	94.7 ± 27.28	125.3 ± 31.72	0.0035
non-HDL-C (mg/dl)	108.3 ± 27.85	145.2 ± 39.88	0.0039
TC/HDL-C ratio	3.06 ± 0.73	4.27 ± 1.06	0.0004
TG/HDL-C ratio	1.42 ± 0.69	2.64 ± 1.60	0.0131
LDL-C/HDL-C ratio	1.80 ± 0.71	2.81 ± 0.80	0.0002

HDL-C: high-density lipoprotein cholesterol; LDL-C: low-density lipoproteins cholesterol; TC: total cholesterol; TG: triglycerides. Values are expressed as mean ± SD. The t-test analysis was performed to compare differences between groups. Statistical significance was considered when *p* < 0.05.

**Table 8 diagnostics-10-01033-t008:** The evolution in lipid profile concerning obese status.

Variable	Non-Obese (*n* = 12)	Overweight (*n* = 9)	Obese I (*n*= 27)	Obese II (*n* = 9)
TG (mg/dl)	75.3 ± 33.84	107.1 ± 64.99	106.8 ± 51.92	139.9 ± 71.71 ^a^
TC (mg/dl)	163.8 ± 25.02	199.0 ± 43.61	190.0 ± 46.67	187.8 ± 40.09
HDL-C (mg/dl)	55.5 ± 10.71	49.0 ± 7.31	47.6 ± 13.01 ^a^	38.7 ± 6.45 ^a,b,c^
LDL-C (mg/dl)	94.7 ± 27.28	126.2 ± 31.57 ^a^	124.1 ± 33.90 ^a^	128.2 ± 28.03 ^a^
non-HDL-C (mg/dl)	108.3 ± 27.85	150.0 ± 44.87 ^a^	142.4 ± 39.96 ^a^	149.0 ± 38.41 ^a^
TC/HDL-C ratio	3.06 ± 0.73	4.15 ± 1.19 ^a^	4.10 ± 0.93 ^a^	4.92 ± 1.16 ^a,c^
TG/HDL-C ratio	1.42 ± 0.69	2.28 ± 1.46	2.41 ± 1.35 ^a^	3.72 ± 2.11 ^a,b,c^
LDL-C/HDL-C ratio	1.80 ± 0.71	2.63 ± 0.84 ^a^	2.68 ± 0.68 ^a^	3.39 ± 0.94 ^a,b,c^

HDL-C: high-density lipoprotein cholesterol; LDL-C: low-density lipoproteins cholesterol; TC: total cholesterol; TG: triglycerides. Values are expressed as mean ± SD. One-way ANOVA was performed to compare differences between groups. Statistical significance was considered when ^a^
*p* < 0.05 vs non-obese; ^b^
*p* < 0.05 vs overweight; ^c^
*p* < 0.05 vs obese I.

**Table 9 diagnostics-10-01033-t009:** Correlation study between body fat percentage (BFP) and waist circumference (WC) vs lipid profile.

	BFP Correlation	WC Correlation
TG (mg/dl)	NS	NS
TC (mg/dl)	r = 0.2790; *p* = 0.0356	NS
HDL-C (mg/dl)	r = −0.2863; *p* = 0.0309	r = −0.4295; *p* = 0.0009
LDL-C (mg/dl)	r = 0.4354; *p* = 0.0007	r = 0.2976; *p* = 0.0246
non-HDL-C (mg/dl)	r = 0.3751; *p* = 0.0040	r = 0.2615; *p* = 0.0494
TC/HDL-C ratio	r = 0.3859; *p* = 0.0030	r = 0.4272; *p* = 0.0009
TG/HDL-C ratio	NS	r = 0.3163; *p* = 0.0165
LDL-C/HDL-C ratio	r = 0.4475; *p* = 0.0005	r = 0.4831; *p* = 0.0001

HDL-C: high-density lipoprotein cholesterol; LDL-C: low-density lipoproteins cholesterol; TC: total cholesterol; TG: triglycerides. Pearson test was performed to evaluate correlations. Statistical significance was considered when *p* < 0.05. NS indicates non-statistical significance.

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
