# Peer review of "Waist Circumference as a Preventive Tool of Atherogenic Dyslipidemia and Obesity-Associated Cardiovascular Risk in Young Adults Males: A Cross-Sectional Pilot Study"

_diagnostics, 2020, doi:10.3390/diagnostics10121033_

Round 1

Reviewer 1 Report

This study shows that waist circumference is a simple and reliable measure to determine the cardiovascular risk in obesity, and could be adopted as an alternative to body mass index to alert obese people at risk of atherogenic dyslipidelimia. 

The artice is written in a properly way, and the data and results are presented and discussed appropriately. So I recommend the publication.

However, I think that some minor changes can improve the quality of the article:

Line 32. The article is written in English, so it would be better the use of Organisation for Economic Cooperation and Development (OECD) instead of the Spanish acronym OCDE.

In the same way, in the line 53 abbreviated phases as IAD or ICCR should be written in full the first time that they appear in the text.

Line 56. “AD is characterized by an elevation TG”. It would be better AD is characterized by an elevation of TG”

Figure 1: Is necessary to improve the quality of the image (it is pixelated). I think the figure must include an explanatory paragraph for better understanding. Visceral adiposity is repeated three time in the pictures, maybe it would be better to change for more specific words, like hepatic fat. It must be specified that TNF is TNF Alpha.

Line 88-96. Were participants from Cordoba, Andalusia or all Spain? It should be mentioned.

Tables 5 and 7. What is the meaning of the Asterisks? It the meaning is statistical significance, it would also appear in tables 4 and 6.

Line 198. Regarding the groups, overweight must be WC between 100 and 110, and obese I WC between 110 and 125.

References format must be carefully revised. In some references like 5, 9, 16, 17 and so on, the name of the journal is not in italics. Also, in reference 54 the complete name of the authors appears before the surname.

Author Response

P.1.

“This study shows that waist circumference is a simple and reliable measure to determine the cardiovascular risk in obesity, and could be adopted as an alternative to body mass index to alert obese people at risk of atherogenic dyslipidelimia. 

The artice is written in a properly way, and the data and results are presented and discussed appropriately. So I recommend the publication.

However, I think that some minor changes can improve the quality of the article:”

Answer P.1. Thank you for your comments and review. We appreciate your time reading this article, and we will encourage to respond to all of your specifications.

P.2.

“Line 32. The article is written in English, so it would be better the use of Organisation for Economic Cooperation and Development (OECD) instead of the Spanish acronym OCDE.”

Answer P.2. True, we apologize for this mistake. Abbreviations must be in English, so we have changed OECD to OECD. Also, we have added the full name of the institution before the acronym. (Line 34).

P.3

“In the same way, in the line 53 abbreviated phases as IAD or ICCR should be written in full the first time that they appear in the text.”

Answer P.3. We thank you again for this comment, which we have corrected. The abbreviations IAS and ICCR have been correctly defined and worded. You can find them now on lines 63-64 of the manuscript.

P.4.

“Line 56. “AD is characterized by an elevation TG”. It would be better AD is characterized by an elevation of TG.”

Answer P.4. Thank you for the grammatical annotation on line 53. We have considered it and changed this paragraph to improve writing. Now, you can read “AD is characterized by an elevation of TG and reduced levels of HDL-C and has been reported to strongly predict CV morbidity, especially coronary artery disease in the general population” (Lines 67-69).

P.5.

“Figure 1: Is necessary to improve the quality of the image (it is pixelated). I think the figure must include an explanatory paragraph for better understanding. Visceral adiposity is repeated three time in the pictures, maybe it would be better to change for more specific words, like hepatic fat. It must be specified that TNF is TNF Alpha.”

Answer P.5. We agree with your comment. We have improved the image quality. Besides, we have added a theoretical explanation of the figure. We have also taken your appreciation into account and modified the names from visceral adiposity to hepatic and muscle. We have corrected the term from TNF to TNF Alpha.

P.6. 

“Line 88-96. Were participants from Cordoba, Andalusia or all Spain? It should be mentioned.”

Answer P.6. We have updated the required information. (See lines 206-208).

P.7.

“Tables 5 and 7. What is the meaning of the Asterisks? It the meaning is statistical significance, it would also appear in tables 4 and 6.”

Answer P.7. We are sorry. You are right. Asterisks indicate significance. It appeared in the final manuscript because of a formatting error by ours. Thus, we have deleted all asterisk from the tables.

P.8.

“Line 198. Regarding the groups, overweight must be WC between 100 and 110, and obese I WC between 110 and 125.”

Answer P.8. We have considered your comment and added the WC range to define obesity subgroups (obese I and obese II) for better understanding. Now, the overweight group is defined as “WC from 100 to 109 cm” instead of WC ≥100 cm. In the same way, the obese I group is defined as “WC from 110 to 124 cm” instead of WC ≥110 cm. (See lines 378-379)

P.9. 

“References format must be carefully revised. In some references like 5, 9, 16, 17 and so on, the name of the journal is not in italics. Also, in reference 54 the complete name of the authors appears before the surname.”

Answer P.9. We are very sorry for the reference format. We have worked hard to adapt the list of references to the correct format and reviewed one by one to adjust the cited number throughout the text.

Reviewer 2 Report

Hernández-Reyes et al. performed a cross-sectional study in which waist circumference could be used as a predictor of atherogenic dyslipidemia and obesity-associated cardiovascular risk in young males.

Although this could be an interesting trial, the main limitation of this study is the sample size analyzed (N=57). In addition, it's a cross-sectional study, thereby it is expected a low impact in its research field.

I suggest authors review and improve manuscript. Design study is incomplert. Some information such as who performs the measurements, how data are collected. Information about hypertension or diabetes is autoreported? how is obtained this information? which is the source? Besides, authors should define all concepts, not all them are defined.

I suggest include some table which use of medications and a descriptive baseline characteritics table (% diabetics, % hypertension, treatment followed, etc.).

I think statistic should be reviewed. Statistics should be adjusted by confusion variables such as hypertension, diabetes or dislipidemia.

Author Response

P.1 “Hernández-Reyes et al. performed a cross-sectional study in which waist circumference could be used as a predictor of atherogenic dyslipidemia and obesity-associated cardiovascular risk in young males.

Although this could be an interesting trial, the main limitation of this study is the sample size analyzed (N=57). In addition, it's a cross-sectional study, thereby it is expected a low impact in its research field.”

Answer P.1. Thanks for reviewing our work. We agree with you that the sample size is small (only 57 men). We define this research as a pilot study (we have added this term in the title) to further investigate the use of WC as a tool to assess the lipid status of patients and assess the CV risk associated with obesity. However, we believe that it is a very homogeneous sample. Only healthy young men were included. We have taken advantage of this comment and extended the reasons for inclusion and exclusion. We are aware that larger sample size, wide age ranges, and a longitudinal evaluation can be the next step in completing this pilot work. In response to your comment, we mention all of these points as limitations (see lines 608-612).

P.2

I suggest authors review and improve manuscript. Design study is incomplert. Some information such as who performs the measurements, how data are collected. Information about hypertension or diabetes is autoreported? how is obtained this information? which is the source?”

Answer P.2. Thank you for your comments. Taking into account the requests of the three reviewers, we have conducted a thorough review of all sections of the manuscript. Materials and methods have been updated. We hope that the revised version provides clarity.

P.3

“Besides, authors should define all concepts, not all them are defined.”

Answer P.3. We appreciate this comment. The concepts have been revised throughout the manuscript. New definitions have been added where appropriate in the original text.

P.4

“I suggest include some table which use of medications and a descriptive baseline characteritics table (% diabetics, % hypertension, treatment followed, etc.).”

Answer P.4. We agree. Any concomitant disease or drugs were exclusion criteria; therefore, the 57 men included in this investigation had no additional health problems. We have improved the inclusion/exclusion criteria subsection to clarify these issues.

P.5

“I think statistic should be reviewed. Statistics should be adjusted by confusion variables such as hypertension, diabetes or dyslipidemia.”

Answer P.5. We are sure that your suggestion could provide useful information on how those variables influence the CV risk evaluation associated with obesity. However, confusion variables were not contemplated in this pilot work because, during the recruitment process, we tried to find a homogeneous healthy population of adult men. So, we cannot perform the suggested statistical analysis. Nevertheless, we will consider this recommendation for future investigation designs.

Reviewer 3 Report

Thank you very much for the opportunity to review this manuscript (ID: diagnostics-994567). I listed my specific considerations below:

TITLE

The title states that WC is a predictor of atherogenic dyslipidemia and risk of CVD. This is confusing because in the following sections of the study I did not find a CVD risk assessment. I only found information about the differences in the mean values of the components of the lipid profile in obese or non-obese young adults males based on BF% and WC or information about the prevalence of the problem under study.

ABSTRACT section

All abbreviations should be explained in this section of the manuscript.

INTRO section

This section is a bit too long. Some of the information is a discussion rather than an introduction. In the lines: 82-85 The Authors claim that they examined the association of WC and BF% with atherogenic dyslipidemia-associated lipid profile and assessed CVD risk. In the following sections of the manuscript I did not find a relationship assessment and risk assessment. However, I found an analysis of differences, prevalence and only one correlation between BF% and WC.

METHODS section

Inclusion/exclusion criteria subsection. I found no mention of what these criteria actually were. In fact, anyone with or without obesity was eligible for this study. Are the Authors sure that the cut-off for BF% for men is 28% ? So far, no clear cut-off criteria have been established for the BF%. The threshold used in this study is too high for this reason. This is confirmed by the results in Table 5. It was easier to calculate this threshold yourself, e.g. based on the ROC analysis.

Statistical analysis subsection. Information on the use of parametric tests is given. Has the normality of the distributions of the variables been examined? What test was used?

RESULTS section

The information contained in tables 2 to 7 can be presented in one summary table. Likewise, Figures 2-4 can be presented as a multi-panel figure. The unit of measure for BMI was not provided. Can 57 men be called a population (Tables 2, 3, also in the text)?

DISCUSSION section

Lines: 345-350. This is a wishful thinking. The longitudinal study (not cross-sectional study) serves the purpose of causal inference. Can the application of a simple 2-compartment model of body composition really be called its exact assessment and a strength of the study?

REFERENCES section

It is a mess here. Some items (e.g. Ross et al.) have been listed multiple times.

EVALUATION

This study has numerous methodological flaws. In defending the thesis, the authors, against the background of an indicator with an unverified cut-off point, try to promote an indicator that has actually been promoted a long time ago. 

Author Response

P.1 

“The title states that WC is a predictor of atherogenic dyslipidemia and risk of CVD. This is confusing because in the following sections of the study I did not find a CVD risk assessment. I only found information about the differences in the mean values of the components of the lipid profile in obese or non-obese young adults males based on BF% and WC or information about the prevalence of the problem under study.”

Answer P.1. We understand your concern about the title statement. To ensure that the title of this research does not affirm incorrect conclusions, we have modified it. This study pretends to elucidate the usefulness of WC in the diagnosis of lipid disruption associated with obesity, which could derive to the future development of atherogenic dyslipidemia and increase the risk of CVD. As we explain in the method subsection “cardiovascular risk assessment” (see lines 267-273), the assessment of CV risk in this work has been performed indirectly by evaluating the prevalence of standardized lipid parameters and ratios for the diagnosis of atherogenic dyslipidemia. We considered the fact that the prevalence is not a rightness tool to test the probability establish any disease. We have included the correlation between body composition and lipid profile and atherogenic ratios in the results section (see lines 446-454)

P.2

ABSTRACT section.

“All abbreviations should be explained in this section of the manuscript.”

Answer P.2. You are right. As a request, all abbreviations have been defined in the abstract section.

P.3

INTRO section.

“This section is a bit too long. Some of the information is a discussion rather than an introduction.”

Answer P.3. We are sorry for the extension of the introduction section. In our opinion, the observational nature of this work should be appropriately justified by given a broader view of the context.

P.4 

“In the lines: 82-85 The Authors claim that they examined the association of WC and BF% with atherogenic dyslipidemia-associated lipid profile and assessed CVD risk. In the following sections of the manuscript I did not find a relationship assessment and risk assessment. However, I found an analysis of differences, prevalence and only one correlation between BF% and WC.”

Answer P.4. Thank you for your comment. In the new manuscript, we have considered the lack of a study in the direct association between body composition and lipid profile, so we have included new data with correlation tests between those parameters in the result section (see line 446-470, new table 9 and new figure 3).

P.5

METHODS section.

“Inclusion/exclusion criteria subsection. I found no mention of what these criteria actually were. In fact, anyone with or without obesity was eligible for this study.”

Answer P.5. Thank you for your comment. We have completed the information about the study design to clarify the inclusion/exclusion criteria.

P.6

“Are the Authors sure that the cut-off for BF% for men is 28% ? So far, no clear cut-off criteria have been established for the BF%. The threshold used in this study is too high for this reason. This is confirmed by the results in Table 5. It was easier to calculate this threshold yourself, e.g. based on the ROC analysis.”

Answer P.6. Thank you for your comment. We agree that this is an important issue and needs to be clarified.

Indeed, 28% of body fat in middle-aged men is an established cut-off point (See Ref 28).

On the other hand, we agree that 28% of body fat should be revised, adapt to the waist circumference measurement in the obesity classification.

P.7

“Statistical analysis subsection. Information on the use of parametric tests is given. Has the normality of the distributions of the variables been examined? What test was used?”

Answer P.7. Thanks for the comment. We have updated this information in the manuscript.(See lines 279-281)

P.8 

RESULTS section

“The information contained in tables 2 to 7 can be presented in one summary table. Likewise, Figures 2-4 can be presented as a multi-panel figure.”

Answer P.8. Although we appreciate your suggestion, we think that one summary table containing data from tables 2 to 7 may be difficult to read. Thus, we have kept data on separate tables. However, we have created a new multi-panel figure involving images 2 to 4 (now Figure 2).

P.9

“The unit of measure for BMI was not provided.”

Answer P.9. We have added the missing units for BMI in tables 2, 4, and 6.

P.10

“Can 57 men be called a population (Tables 2, 3, also in the text)?”

Answer P.10. We agree with you that “population” maybe no the best term to define the number of participants in this work. Thus, we switched the word “population” to “sample”

P.11

DISCUSSION section

“Lines: 345-350. This is a wishful thinking. The longitudinal study (not cross-sectional study) serves the purpose of causal inference. Can the application of a simple 2-compartment model of body composition really be called its exact assessment and a strength of the study?”

Answer P.11. We apologize for the lack of clarity in this paragraph. We believe that a simple model that is easy to measure and that can serve as a preventive tool for cardiovascular disease can be a strength. But also, we recognize the scant forcefulness of such a statement. After careful thought, we have decided to remove this paragraph from the manuscript.

P.12. 

REFERENCES section

“It is a mess here. Some items (e.g. Ross et al.) have been listed multiple times.”

Answer P.12. We are very sorry for the reference format. We have worked hard to adapt the list of references to the correct format and reviewed one by one to adjust them throughout the text. We have also removed those duplicate references.

P.13

EVALUATION

“This study has numerous methodological flaws. In defending the thesis, the authors, against the background of an indicator with an unverified cut-off point, try to promote an indicator that has actually been promoted a long time ago.”

Answer P.13. Thank you for your exhaustive review of our work. We apologize for the errors committed. We have worked on improving this manuscript. We have completed or modified each section to facilitate the transmission of the results and conclusions of this work. We want to emphasize the main aim of this work, which was not to standardized the measure of WC (that is already widely in use) but also to use it as a complementary tool when assessing patients about their CV status.

Round 2

Reviewer 2 Report

Authors have improved their manuscript, however some points should be reviewed.

An extensive editing of English language and style required.

Change this sentece "All data collected in the study sample was obtain by an experienced nutritionist" by "All study data collected werer obtained by a trained an experimented dietitian". Delete the name of the dietitian. I think this inforamtion is not necessary.

Parameters related with physical activity how are collected? usig a device? using a validated test? Nutritional data are collected? (FFQ, 24h-record, etc.)...diet could influence in data results.

Authors say that during study differents variables are collected through different face-to face interview...can you define how many?

Is a requisit to have a device to collect physical activity data? is this a motive of exclusion criteria? (clarify please)

Author Response

P.1 Authors have improved their manuscript, however some points should be reviewed.

Answer P.1

Dear reviewer,

We appreciate your comments. We have tackled each question with interest and explain point by point.

P.2 An extensive editing of English language and style required.

Answer P.2

We understand that we can improve the quality of the language edition. We have thoroughly reviewed the text. We hope you find it correct this time. We are at your disposal for any aspect related to the language.

P.3 Change this sentece "All data collected in the study sample was obtain by an experienced nutritionist" by "All study data collected werer obtained by a trained an experimented dietitian". Delete the name of the dietitian. I think this inforamtion is not necessary.

Answer P.3

We appreciate this comment. This information has been requested on some occasions, but we agree with you, it is not necessary, and we have removed it from the text

P.4 Parameters related with physical activity how are collected? usig a device? using a validated test? Nutritional data are collected? (FFQ, 24h-record, etc.)...diet could influence in data results.

Answer P.4

Thanks for this comment. This comment on materials and methods can lead to confusion. Let us remember that this is an observational study; we have not previously manipulated any patient variable through diet or prescription of physical activity. We have removed this phrase.

P.5 Authors say that during study differents variables are collected through different face-to face interview...can you define how many?

Answer P.5

We feel the confusion. There is only one individual interview with each patient, in which blood is extracted for biochemical analysis, and the corresponding anthropometric records are given (height, age, and body composition). We have reviewed the methodology, and we believe the information is clear. If this is not the case, please tell us where it finds the discrepancy.

P.6 Is a requisit to have a device to collect physical activity data? is this a motive of exclusion criteria? (clarify please)

Answer P.6

As we have previously commented, in this observational study, we try to understand the relationship between body composition (percentage of body fat, waist circumference) and the risk of atherogenic dyslipidemia through the primary lipid markers. The sample is sedentary and overweight men. We have eliminated the mention of physical activity data from the methodology

Reviewer 3 Report

I appreciate the corrections made. Thank you.
However, I renew my request to the Authors. Please calculate the BF% cut-off value for this group. I suggest. Statistical method - ROC analysis. Predictive variable - BF%. Binary classifier - dyslipidemia (yes or no). Optimal cut-off value - specific point on the curve whereupon Youden index (defined as sensitivity + specificity - 1) is at the maximum. This is a very simple analysis and your research will be credible. Why do I think so? See Table 2. Based on BMI, this group was obese and non-obese based on BF%. Table 4. Non-obese group mean value based on BF% = 21.1 (high standard error) and obese group BF% = 32.2 (small standard error). I think the cut-off for this group is around 25%.

My last comment concerns Figure 2. Using the statistical test, please compare the presented proportions. For Figure 2C (four categories), Bonferroni correction for P value should be used.

Author Response

P.1 I appreciate the corrections made. Thank you.

Answer P.1

We appreciate this feedback. We agree with your comment and have expanded the statistical test. Hopefully, we improved the review.

P.2 However, I renew my request to the Authors. Please calculate the BF% cut-off value for this group. I suggest. Statistical method - ROC analysis. Predictive variable - BF%. Binary classifier - dyslipidemia (yes or no). Optimal cut-off value - specific point on the curve whereupon Youden index (defined as sensitivity + specificity - 1) is at the maximum. This is a very simple analysis and your research will be credible. Why do I think so? See Table 2. Based on BMI, this group was obese and non-obese based on BF%. Table 4. Non-obese group mean value based on BF% = 21.1 (high standard error) and obese group BF% = 32.2 (small standard error). I think the cut-off for this group is around 25%.

Answer P.2

Effectively. As predicted, if we perform this test, the proposed cut-off point is 23.4% fat percentage. This is close to the 25% who suspected. We have included the following sentence in the discussion: "A new optimal cut-off point should be considered to determine the percentage of body fat that best classifies the degree of obesity in adult men. Considering our data, with a ROC analysis, we found this point at a value of 23.4% of total body fat "(see lines 578-580)

P.3 My last comment concerns Figure 2. Using the statistical test, please compare the presented proportions. For Figure 2C (four categories), Bonferroni correction for P value should be used.

Answer P.3

Dear reviewer,
We have done the test that indicates that we do not see any change in the figure. With this test we simply show information about percentages and found risk.
If you understand that this figure should not be in the manuscript, we can remove it.

It is the only point that we do not see clear how to proceed. We are sorry.

Round 3

Reviewer 2 Report

Thanks for your comments. For me that is all

Author Response

Dear reviewer,

We sincerely appreciate your feedback during the review process. we are convinced that the manuscript has improved remarkably thanks to your comments.

Again, thank you very much
Sincerely

Reviewer 3 Report

Dear Author’s
Thank you for considering my suggestion for the ROC analysis. The second suggestion was only to compare these proportions, e.g. using this test: https://www.medcalc.org/calc/comparison_of_proportions.php. This will allow to determine whether the differences are statistically significant or not. In the case of Figure 2C, each studied variable has 4 categories. Therefore, when comparing groups in pairs (1 to 2; 1 to 3, 1 to 4, etc.), the nominal significance level should be reduced. In this particular case it will be 0.01 (alpha = 1 - 0.95 ^ 0.25 or just alpha = 0.05 / 4). In my opinion, the suggested change will only improve the interpretation of the results, but will not change them. I strongly advise against deleting Figure 2. This figure is good.

Author Response

Dear reviewer,
We sincerely appreciate your feedback during the review process. We are convinced that the manuscript has improved thanks to your comments remarkably.
As requested, we have performed the statistical test and compared the groups in the figure.
We have also updated the information on materials and methods.
Please, if you detect an anomaly, we will be happy to correct it.

Again, thank you very much.
Sincerely